Screening and verification of proteins that interact with the anthocyanin-related transcription factor PbrMYB114 in ‘Yuluxiang’ pear

Liu Qingwei
Gao Ge
Shang Chen
Li Tong
Wang Yadong
Li Liulin sxaulll@sxau.edu.cn
Feng Xinxin fengxx@sxau.edu.cn
College of Horticulture, Shanxi Agricultural University , Taigu, Shanxi Province , China
Sotelo-Mundo Rogerio
Electronic publication date: 2024 Jun 14
Publication date: 2024
Volume: 12
Electronic Location ID: e17540
Received 2024 Feb 14; Accepted 2024 May 19
Copyright: © 2024 Liu et al.
Copyright year: 2024
Copyright holder: Liu et al.
License: This is an open access article distributed under the terms of the Creative Commons Attribution License, which permits unrestricted use, distribution, reproduction and adaptation in any medium and for any purpose provided that it is properly attributed. For attribution, the original author(s), title, publication source (PeerJ) and either DOI or URL of the article must be cited.
License URL: https://creativecommons.org/licenses/by/4.0/

Keywords: Yuluxiang pear, Anthocyanin, PbrMYB114, Interacting proteins, Metallothioneins

Funding: Shanxi Province Science and Technology 202201140601027-2 Science and Technology Innovation Fund Project of Shanxi Agricultural University 2017YJ45 Fundamental Research Program of Shanxi Province 202303021211097 This work was supported by the Shanxi Province Science and Technology Major Special Project (Grant No. 202201140601027-2), the Science and Technology Innovation Fund Project of Shanxi Agricultural University (Grant No. 2017YJ45) and the Fundamental Research Program of Shanxi Province (Grant No. 202303021211097) The funders had no role in study design, data collection and analysis, decision to publish, or preparation of the manuscript.

==============================
Despite extensive research highlighting the pivotal role of MYB transcription factors in regulating anthocyanin biosynthesis, the interactive regulatory network involving these MYB factors in pear fruits remains inadequately characterized. In this study, the anthocyanin-regulatory gene PbrMYB114 was successfully cloned from ‘Yuluxiang’ pear (Pyrus bretschneideri) fruits, and its influence on anthocyanin accumulation was confirmed through transient expression assays. Specifically, the co-transformation of PbrMYB114 with its partner PbrbHLH3 in pears served to validate the functional role of PbrMYB114. Subsequently, PbrMYB114 was employed as bait in a yeast two-hybrid screening assay, using a ‘Yuluxiang’ pear protein library, which led to the identification of 25 interacting proteins. Further validation of the interactions between PbrMYB114 and PbrMT2/PbrMT3 was conducted. Investigations into the role of PbrMT2 and PbrMT3 in ‘Duli’ seedlings (Pyrus betulaefolia) revealed their potential to enhance anthocyanin accumulation. The outcomes of these studies provide novel insights into the protein network that regulates pear anthocyanin biosynthesis, particularly the functional interactions among PbrMYB114 and associated proteins.

Introduction

Pears (Pyrus) are among the most popular fruits in the world, with red-skinned pears in particular being widely consumed owing to their bright appearance and perceived nutritional benefits (Bai et al., 2017). The manifestation of the red coloration in these pears arises from the accumulation of anthocyanins, which are important regulators of fruit ripening and quality (Wang et al., 2018a). The consumption of anthocyanin-rich foods is increasingly recognized for its health benefits, including the prevention of cardiovascular diseases and enhancement of anti-inflammatory responses due to these flavonoids’ potent antioxidant activities (Salehi et al., 2020; Tsuda, 2012). Anthocyanins also contribute to plant resilience against environmental stress and attract pollinators (Shang et al., 2011). Despite making significant advancements in studying the structures of anthocyanins, there are still many parts of the anthocyanin biosynthesis regulatory network that have not been identified (Liu et al., 2021).

The synthesis of anthocyanins, a class of pigments, is facilitated by the phenylpropanoid metabolic pathway, wherein a set of transcription factors (TFs) may play a noticeable role in regulatory coordination. Specifically, TFs belonging to the MYB family exert control over the expression of various structural genes, thereby playing a key role in the intricate regulation of anthocyanin biosynthesis in plants (Feller et al., 2011). In Arabidopsis, for example, the MYB TFs AtMYB75, AtMYB90, AtMYB113, and AtMYB114 have been demonstrated to regulate the synthesis of anthocyanins (Gonzalez et al., 2008), and more recent studies have documented the ability of growing numbers of MYB TFs to influence fruit coloration in various fruit trees (Espley et al., 2007; Feng et al., 2015b). In apples, the transcription factors MdMYB1 and MdMYB10 are recognized members of the MYB family, playing pivotal roles in the modulation of anthocyanin content (Ban et al., 2007). Similarly, in pears, various MYBs have been characterized for their regulatory influence on anthocyanin levels. PyMYB10 and PyMYB10.1 have been pinpointed as key orchestrators of anthocyanin biosynthesis (Feng et al., 2015a). Additionally, PpMYB114 has been identified as a potent activator that significantly boosts anthocyanin production (Yao et al., 2017). The ‘Yuluxiang’ pears (Pyrus bretschneideri) represent a highly favored cultivar in China, yet they are distinguished by their peel’s suboptimal coloration. Consequently, investigations into the MYB-mediated regulatory network, which is responsible for the synthesis and accumulation of anthocyanins in ‘Yuluxiang’ pears, could significantly contribute to the enhancement of pear quality and offer valuable insights into the underlying biological processes.

Plant MYB transcription factors (TFs) typically regulate anthocyanin biosynthesis through interactions with basic helix-loop-helix (bHLH) and WD repeat (WDR) proteins, which together form trimeric MYB–bHLH–WDR (MBW) protein complexes (Tao et al., 2018). bHLH TFs are essential for R2R3-MYB activity owing to their ability to promote their transcription and/or to stabilize MYB complexes (Yan et al., 2020). For instance, apple skin pigmentation is governed by the interaction between MdMYB1 and MdbHLH3, which positively regulates anthocyanin biosynthesis (Ban et al., 2007). Similarly, pear MYB proteins such as PyMYB114, PyMYB10, and PyMYB10b interact with PybHLH3 to significantly enhance the production of anthocyanins (Zhai et al., 2016). Moreover, PpMYB140 has been shown to repress anthocyanin synthesis in red pear fruit by competitively binding to PpbHLH3 (Ni et al., 2021). Ethylene response factors (ERFs) also participate in anthocyanin regulation by interacting with MYB TFs (Sun et al., 2023). In pears, PpMYB114 interacts with ERFs like PpERF24 and PpERF96, which in turn strengthens its interaction with PpbHLH3 and promotes anthocyanin synthesis (Ni et al., 2019). The ability of MYBs to interact with other proteins is essential for the regulation of anthocyanin biosynthesis (Li et al., 2022a). Several anthocyanin repressors, including a diverse array of proteins, have been identified; these repressors predominantly suppress anthocyanin biosynthesis by inhibiting the transcription of MBW complex components or by destabilizing these complexes through protein-protein interactions (LaFountain & Yuan, 2021). For example, the apple RING E3 ubiquitin ligase MdMIEL1, known as MYB30-INTERACTING E3 LIGASE 1, suppresses anthocyanin accumulation by interacting with MdMYB1 (An et al., 2017). Additionally, CONSTITUTIVELY PHOTOMORPHOGENIC 1 (COP1) has been shown to negatively affect light-induced anthocyanin biosynthesis (Podolec & Ulm, 2018), and interactions between MdMYB1 and MdCOP1 are known to impact fruit coloration (Li et al., 2012).

The yeast two-hybrid (Y2H) system is a valuable tool for swiftly detecting novel protein-protein interactions across various organisms (Rodriguez-Negrete, Bejarano & Castillo, 2014). This approach has been effectively utilized to discover potential interactors for numerous key plant proteins (Kong et al., 2019; Wang et al., 2022, 2021b, 2021c). For instance, a Y2H screen using an apple cDNA library identified MdBT2, a nitrate-responsive protein interacting with MdMYB1 and influencing anthocyanin accumulation (Wang et al., 2018b). Similarly, the Y2H method clarified the interaction between MdWD40-280 and MdMYB110a, which is relevant to anthocyanin accumulation (Fang, Shangguan & Wang, 2023). These findings underscore the significance of protein-protein interactions in the regulation of anthocyanin biosynthesis by MYB transcription factors (TFs).

In this study, we confirmed the functional significance of the PbrMYB114 gene, which is associated with anthocyanin regulation in ‘Yuluxiang’ pears, using transient transformation experiments. Subsequently, a yeast two-hybrid (Y2H) screening was employed to identify proteins interacting with PbrMYB114. By studying the interaction between MYB and other regulatory factors, the regulatory network of PbrMYB114 involved in anthocyanin metabolism was preliminarily revealed. The characterization of these interactors is expected to shed light on the molecular mechanisms governing fruit color development in pears.

Materials and Methods

Plant materials and growth

‘Yuluxiang’ pears (Pyrus bretschneideri) were obtained 135 days after flower blossom in 2021 from an orchard in Xi County, Linfen City, Shanxi province, China (coordinates: 36°41′39.5″N, 110°55′46.8″E). In order to obtain a representative sample, 5–10 fruits from different trees were combined to form one sample. A composite library was constructed using peel samples exhibiting varying colors—red, green, and yellow—from these pears. Specifically, unpigmented ‘Yuluxiang’ pears fruits, and ‘Duli’ pear seedlings (Pyrus betulaefolia) with 2 to 3 leaves were chosen for transient expression assays. Leaves and fruits were first cut into smaller pieces, then snap-frozen in liquid nitrogen, and finally stored at −80 °C for subsequent analysis.

Sequence evolution analyses

The PbrMYB114 protein sequences underwent analysis using the Basic Local Alignment Search Tool for proteins (BLASTP) to identify homologous sequences. A Neighbor-Joining phylogenetic tree was subsequently constructed with MEGA 7.0, bolstered by 1,000 bootstrap replicates to ensure statistical robustness.

cDNA expression library construction and evaluation

TRIzol (Vazyme, Nanjing, China) was used to isolate total RNA from the peels of three different colors of ‘Yuluxiang’ pears and confirmed with 1% agarose gel electrophoresis and a nucleic acid analyzer (Thermo Nanodrop; Thermo Fisher, Waltham, MA, USA) to assess RNA integrity and purity. Then, oligo (dT) magnetic beads were utilized to purify mRNA from samples for the construction of a cDNA library. The purified mRNA was used with the SMARTTM cDNA synthesis technology (Clontech, San Jose, CA, USA) to prepare cDNA for library construction. As directed by the manufacturer, cDNA was normalized to eliminate any low molecular weight cDNA fragments or small pieces of contaminating DNA (Bogdanova et al., 2009) followed by purification using a CHROMA SPIN-1000 chromatographic column (Clontech, San Jose, CA, USA). The resultant cDNA was then recombined with pGAD-T7 and electroporated into TOP10 to generate the library, followed by the culture of dilutions of the mixed transformed population (from 1:10 to 1:1,000) on solid LB medium containing 100 mg·L−1 ampicillin, assessing colony numbers and transformation efficiency after 12 h. In total, 24 clones were selected at random and analyzed by PCR with universal pGAD-T7 -F/R primers (Table S1) to assess insert sizes and Y2H library recombination rates via 1% agarose gel electrophoresis.

Bait plasmid auto-activation and toxicity analyses

Prior to Y2H screening, the full-length PbrMYB114 coding sequence was cloned with the Phanta Max Super-Fidelity DNA Polymerase (Vazyme, Nanjing, China) using gene-specific primers and introduced into the pGBKT7 vector using the EcoRI and SalI sites (Takara, Kusatsu, Japan) using Exnase II (Vazyme, Nanjing, China). AH109 cells were then separately transformed with the pGBKT7-PbrMYB114 and pGBKT7 plasmids, and transformants were grown for 3–5 days on SD/-Trp medium at 30 °C. The auto-activation ability of the PbrMYB114 bait was tested by transforming yeast competent AH109 cells with the pGBKT7-PbrMYB114 bait vector and the empty pGADT7 vector followed by selection on SD/-Trp/-His, SD/-Trp/-His/-Ade, SD/-Trp/-His/-Ade + X-α-gal medium.

PbrMYB114 interacting protein screening

To screen for PbrMYB114-interacting proteins, the pGBKT7-cDNA library plasmid was transformed into the AH109 yeast strain containing the pGBKT7-PbrMYB114 bait plasmid as a receptor to generate competent cells that were plated for 3–5 days on SD/-Trp/-Leu/-His/ + 30 mM 3-AT (3-amino-1, 2, 4-triazole) medium at 30 °C Colonies >2 mm in size were selected for additional analysis on SD/-Ade/-His/-Leu/-Trp/X-α-Gal/AbA + 30 mM 3-AT plates for 3–5 days. Genes in positive clones were identified by amplifying DNA from these clones with a rapid amplification yeast positive clone kit (RY8001; ProNet Biotech, China) followed by DNA sequencing and BLAST analysis against the Chinese white pear genome (http://peargenome.njau.edu.cn, accessed on 23 February 2022).

Validation of putative PbrMYB114 interactors

For verification of candidate protein-protein interactions discerned through Y2H screening, Y2HGold yeast cells underwent transformation, wherein each cell was individually introduced with the pGBKT7-PbrMYB114 bait plasmid, in conjunction with distinct prey plasmids encapsulating the sequences of putative interacting proteins. Transformants were cultured for 3–5 days on non-selective double drop-out (DDO, SD/-Leu/-Trp), selective drop-out (TDO, SD/-Leu/-Trp/-His), and quadruple drop-out (QDO, SD/-Leu/-Trp/-His/-Ade) medium, adding 20 mg/L X-α-galactose (X-α-gal) to the QDO medium to detect α-galactosidase activation as a result of protein-protein interactions, with blue colonies being indicative of true positive interactions. Empty vector transformation was performed as a control.

Total anthocyanin extraction and quantification

Anthocyanins were extracted following the protocol established by Ubi et al. (2006). Approximately 0.1 g of ‘Yuluxiang’ pear fruits skin and ‘Duli’ seedling leaves were submerged in 2 ml of cold methanol containing 0.1% HCl and kept in the dark at 4 °C for 24 h. The absorbance of the extracted samples was measured using an Eppendorf spectrophotometer (Eppendorf AG, Hamburg, Germany) at wavelengths of 530, 620, and 650 nm.

Transient transformation of pear fruits and pear leaves

It was attempted to amplify PbrMYB114, PbrbHLH3, PbrMT2, and PbrMT3 coding sequences from cDNA that was prepared from the skin of ‘Yuluxiang’ pears using appropriate primers (Table S1) and introduced into the pNC-Cam2304-MCS35S vector via Nimble Cloning. Agrobacterium EHA105 cells were then transformed with these recombinant constructs and incubated for 2 days at 28 °C, using the empty vector as a control. For transient overexpression performed as reported previously, these Agrobacterium cells were suspended in infiltration buffer (10 mM MgCl2, 10 mM MES, pH 5.6, 200 mM acetosyringone) at a final OD600 of one and cultivated for 3 h at 25 °C at 70 rpm before injection. The Agrobacterium mixture was injected into pear fruit, five pears per gene and, control was used for each treatment. ‘Yuluxiang’ pear fruits underwent transient transfection with PbrMYB114 either individually or in conjunction with PbrbHLH3. The pear fruit was kept in the dark overnight, and then exposed to normal outdoor light for 5 days to observe the phenotype. The transient transformation of ‘Duli’ seedlings (Pyrus betulaefolia) with 2–3 leaves was performed by the Agrobacterium cells transformed with the recombinant vectors. Each ‘Duli’ seedlings comprised three biological replicates, and each replicate included three. Then followed by infiltration for 20 min using a GM-0.33A vacuum pump (Jinteng, Wuhan City, China). After infection, ‘Duli’ seedlings were incubated in an artificial incubator (GZL-P160-B2; Anhui, China) in the dark for 24 h before incubation at 24 °C under a light source with a 16 h daylight period. At 10 days post-infection, leaves were photographed and used for RNA extraction and measurement of anthocyanin content.

RNA extraction and qPCR

The CTAB-LiCl method, enhanced for efficacy, was employed to extract total RNA from the designated samples (Zhao et al., 2012). Designing primers tailored for PbrMYB114, PbrbHLH3, and the reference gene Pbractin was undertaken via Primer Premier 5.0 (Table S1). Subsequent qPCR analysis was carried out utilizing a SYBR Premix Ex Taq II kit from Vazyme and a Light Cycler 96 Real-Time PCR System (Roche LightCycler, Shanghai, China) that was headquartered in Switzerland, with three replicates per experiment. The relative expression of the genes was calculated using the 2−∆∆CT method (Livak & Schmittgen, 2001). Three biological replicates were used for all analyses and error bars.

Data analysis

Statistical analyses were performed by Duncan’s test using SPSS26. Different lowercase letters indicate significant differences between different treatment samples (p < 0.05). while Figures were generated with GraphPad Prism 9.0.

Results

PbrMYB114 cloning

The PbrMYB114 target fragments were amplified via PCR and fragment size were evaluated by 1% agarose gel electrophoresis (Fig. 1), revealing that the PbrMYB114 cDNA was 690 bp in length. Neighbor-joining phylogenetic analyses of the amino acid sequence encoded by PbrMYB114 and related sequences in other species indicated that PbrMYB114 exhibited the strongest similarity with the Pyrus bretschneideri PyMYB114 and Pyrus pyrifolia PyMYB10.1.

Figure 1 PbrMYB114 identification and characterization.

(A) The PbrMYB114 cDNA fragment. (B) Phylogenetic analysis of the amino acid sequence encoded by PbrMYB114.

Evaluation of the role of PbrMYB114 as a regulator of pear anthocyanin biosynthesis

Transient transformation in ‘Yuluxiang’ pear fruits was performed to test the function of the PbrMYB114 in the context of anthocyanin production. No pigmentation was evident in fruits transformed with PbrMYB114 or PbrbHLH3 alone or with the empty vector control (Fig. 2), whereas a pronounced enhancement in pigmentation was evident following PbrMYB114 and PbrbHLH3 co-transformation. Changes in anthocyanin levels in these ‘Yuluxiang’ pears were consistent with these visual results (Fig. 2B). We overexpressed both PbrMYB114 and PbrbHLH3 in pear and detected the increased expression of both genes by RT-PCR (Fig. 2C). In conclusion, the co-transformation of PbrMYB114 and PbrbHLH3 significantly enhanced anthocyanin biosynthesis, as evidenced by the increased expression levels of both genes and the observed changes in pigmentation in ‘Yuluxiang’ pear fruits.

Figure 2 Evaluation of the eûects of overexpressing PbrMYB114 in pears. Different lowercase letters indicate significant differences between different treatment samples (p < 0.05).

(A) The appearance of the indicated transformed pears on day 5 post-injection. (B) Anthocyanin levels in peels from pears in the indicated group following light treatment. (C) PbrMYB114 and PbrbHLH3 expression in peels from pears following light treatment. Error bars indicate means ± SD (n = 3). Different letters on bars indicate significant differences determined at the level of p < 0.05 by Turkey’s post hoc text.

Construction of a yeast two-hybrid library

To facilitate the reliable identification of functional protein-protein interactions, constructing a high-quality Y2H cDNA library is essential. Extracted RNA and established library quality was first validated. The quality of the total RNA samples shown in Fig. 3A, revealed the expected 28S and 18S ribosomal bands, whereas corresponding 5S bands were largely absent, consistent with a lack of any substantial RNA degradation. The normalization showed that dscDNAs were uniformly dispersed, revealing a range of cDNA fragments from 0.5 to 5 kb in size (Fig. 3B). The resultant Y2H cDNA library included 1.592 × 103 independent clones with a titer of 7.96 × 10⁷ cfu/ml (Fig. 3C). Analyses of 24 colonies selected at random revealed library insert sizes from 500 to 2,000 bp (Fig. 3D). These findings demonstrated that this cDNA library was of acceptable quality such that it was suitable for screening for PbrMYB114-interacting proteins.

Figure 3 Y2H library construction and quality analyses.

(A) Evaluation of the integrity of total RNA from ‘Yuluxiang’ pears. (B) The ds cDNA after normalization and purifcation were evaluated by 1% agarose gel electrophoresis. (C) Library cell density evaluation, plate counting of 1,000-fold diluted yeast cells from the yeast library (D) PCR based assessment of insertion fragments (Lane 2–24). M: DL DL5000 marker; 1: RNA or cDNA from ‘Yuluxiang’ pears.

Bait plasmid auto-activation and toxicity analyses

In an effort to detect potential auto-activation activity, the AH109 yeast strain was transformed with the pGBKT7-PbrMYB114 bait plasmid and subsequently cultured for 3–5 days on SD/-Trp, SD/-Trp/-His, SD/-Trp/-His/-Ade, and SD/-Trp/-His/-Ade+X-α-gal agar plates, with pGBKT7 transformation serving as a negative control. Yeast transformed with the recombinant pGBKT7-PbrMYB114 plasmid were able to grow on SD/-Trp agar plates (Fig. 4), demonstrating successful host cell transformation without substantial toxicity. Negative control cells only grew on SD/-Trp plates, and colonies grown on SD/-Trp/-His/-Ade+X-α-gal plates were not blue. In contrast, pGBKT7-PbrMYB114 colonies grown on SD/-Trp/-His, SD/-Trp/-His/-Ade plates confirmed reporter gene activation in yeast transformed with pGBKT7-PbrMYB114, consistent with the self-activating ability of pGBKT7-PbrMYB114. To assess the most appropriate 3-AT concentration to inhibit this auto-activation, yeast transformed with pGBKT7-PbrMYB114 and pGBKT7 were cultured in SD/-Trp/-His deficient medium containing a range of 3-AT concentrations. These analyses revealed the ability of 30 mM 3-AT to inhibit bait plasmid self-activation such that this system was considered suitable for further Y2H screening.

Figure 4 Analyses of pGBKT7-PbrMYB114 bait vector toxicity and auto-activation.

(A) Verification of the self-activation ability of PbrMYB114. (B) The inhibition of PbrMYB114/pGBKT7 auto-activation on SD/-Trp/-His medium in the presence of the indicated 3-AT concentrations.

PbrMYB114 interactor screening

To screen for proteins capable of interacting with PbrMYB114, the ‘Yuluxiang’ pear Y2H library was screened using pGBKT7-PbrMYB114 as the bait. Those transformants exhibiting interactions between pGBKT7-PbrMYB114 and AD library proteins in Y2H Gold yeast cells were incubated on SD/-Trp/-Leu/-His containing 30 Mm 3-AT, and 96 single colonies were transferred onto SD/-Trp/-Leu/-His for further screening. Following positive clone sequencing, a total of 25 positive clones were obtained (Table 1 and Fig. S2), consisting primarily of metallothionein-like and auxin-responsive proteins that may function through interactions with PbrMYB114.

Table 1 Putative proteins that interact with PbrMYB114 identified through Y2H library screening.

Gene number	Gene name	Accession numbers	Predicted protein	
1	Pbr029632	LOC103947422	60S ribosomal protein L18a-2-like	
2	Pbr037168	LOC103936035	Eukaryotic translation initiation factor 3 subunit J-like	
3	Pbr015437	LOC103961836	Cytochrome P450 CYP736A12-like	
4	Pbr024041	LOC103926780	Uncharacterized LOC103926780	
5	Pbr000210	LOC103956260	Oxygen-evolving enhancer protein 3-2, chloroplastic	
6	Pbr033973	LOC103948939	60S ribosomal protein L35-like	
7	Pbr041033	LOC103951972	Photosystem II 10 kDa polypeptide, chloroplastic-like	
8	Pbr026185	LOC103928700	3-ketoacyl-CoA thiolase 2, peroxisomal-like	
9	Pbr040207	LOC103937799	RING-H2 finger protein ATL43-like	
10	Pbr020842	LOC103944950	Uncharacterized LOC103944950	
11	Pbr000392	LOC103934171	Auxin-repressed 12.5 kDa protein-like	
12	Pbr024120	LOC125471082	Uncharacterized protein	
13	Pbr006261	LOC103941805	Metallothionein-like protein type 2	
14	Pbr000365	LOC103932067	Uncharacterized LOC103932067	
15	Pbr018676	LOC103964341	Protein SGT1 homolog	
16	Pbr042213	LOC103939008	Auxin-responsive protein SAUR21-like	
17	Pbr009476	LOC103936339	Sedoheptulose-1,7-bisphosphatase, chloroplastic-like	
18	Pbr013264	LOC103947227	GDT1-like protein 4	
19	Pbr005665	LOC103955273	Methyl-CpG-binding domain-containing protein 11-like	
20	Pbr004396	LOC103954436	40S ribosomal protein S24-1	
21	Pbr017670		Uncharacterized protein	
22	Pbr000755	LOC103964652	Mitochondrial dicarboxylate/tricarboxylate transporter DTC	
23	Pbr036394	LOC103935622	Poly(ADP-ribose) glycohydrolase 1-like	
24	Pbr034880	LOC103953524	Protein NUCLEAR FUSION DEFECTIVE 6, chloroplastic/mitochondrial	
25	Pbr017424	LOC103947618	Metallothionein-like protein type 3	

Probing protein interactions with PbrMYB114 through yeast two-hybrid analysis

To validate the above Y2H screening results, two metallothioneins (MTs) proteins were selected from among the 25 putative interactors for further evaluation. Yeast cells, co-expressing PbrMYB114 alongside each of the 25 interactors individually, exhibited growth on the selective triple drop-out (TDO) medium, indicating consistent positive regulatory interactions, as illustrated in Fig. S1. The full-length coding sequences for PbrMT2 and PbrMT3 were amplified and cloned into the pGADT7, after which Y2HGold yeast were co-transformed with these constructs and the pGBKT7-PbrMYB114 plasmid. The resultant cells were plated on DDO, QDO/X/3AT media using a gradient dilution approach, revealing that cells expressing these pairs of proteins grew normally on DDO (SD/-Trp/-Leu) medium and produced blue colonies on QDO/A/X/3-AT (SD/-Trp/-Leu/-His/-Ade/X-α-gal/3-AT) medium (Fig. 5). These findings provided further validation for the observed Y2H library screening results.

Figure 5 Yeast two-hybrid assay demonstrating interactions between PbrMYB114 and putative interactors.

Co-expression of pGBKT7-PbrMYB114 and pGADT7(AD)-interactors occurred in yeast on DDO (-Leu/-Trp) medium and QDO medium supplemented with 20 mg/L X-α-gal. Control experiments involved the use of empty vectors.

Transient co-transformation of ‘Duli’ seedlings with candidate proteins and the PbrMYB114–PbrbHLH3 modulates anthocyanin biosynthesis

To explore the mechanisms whereby these putative interacting proteins may influence anthocyanin biosynthesis in pears, ‘Duli’ seedlings were transiently transformed with PbrMT2, PbrMT3, and related TFs (PbrMYB114 and PbrbHLH3) known to serve as anthocyanin pathway activators. While empty vector or PbrMT3 injection alone failed to alter pigmentation (Fig. 6), some pigmentation was evident following PbrMYB114 and PbrbHLH3 co-transformation, and substantial changes were observed following co-transformation with PbrMT2 or PbrMT3 together with PbrMYB114 and PbrbHLH3. Strikingly, the co-transformation of leaves with PbrMYB114/PbrbHLH3 and PbrMT3 resulted in more pronounced pigmentation than PbrMT2 co-transformation. A significant change in total anthocyanin content was also observed (p < 0.05), consistent with the observed phenotypic changes in ‘Duli’ seedlings (Fig. 6B). These findings demonstrate that PbrMT2 and PbrMT3 not only serve as regulators of anthocyanin production, but they may also work by interacting with the PbrMYB114 and PbrbHLH3 complex to regulate anthocyanin levels.

Figure 6 Analyses of the effects of the PbrMT2 and PbrMT3 co-transformation of pear leaves together with interacting partners on anthocyanin biosynthesis.

(A) Pear leaf phenotypic changes on day 10 post-infiltration. (B) Total anthocyanin levels in pear leaves. Error bars indicate means ± SD (n = 3). Different letters on bars indicate significant differences determined at the level of p < 0.05 by Turkey’s post hoc text.

Discussion

Anthocyanin is a flavonoid secondary metabolite, and MYB family TFs have been firmly established as key regulators of anthocyanin biosynthesis. In Asian pears, PyMYB10 was reported to be correlated with the production of anthocyanins (Feng et al., 2010), yet mapping data from European pears (Pyrus communis L.) indicated that PcMYB10 does not serve as a direct regulator of red skin coloration in this variety (Pierantoni et al., 2010). Different MYB TFs may thus play cultivar-specific roles in regulating the anthocyanin pathway in pear fruits. Here, the functional roles of PbrMYB114 and its partner PbrbHLH3 were confirmed in ‘Yuluxiang’ pears via transient transformation (Fig. 2), yielding results in line with those published by Yao et al. (2017).

Y2H library screening was performed for PbrMYB114, leading to the identification of 25 independent interacting proteins, including a RING-H2 finger ATL43-like protein. RING finger proteins exhibit E3 ubiquitin ligase activity and can participate in light signal pathway (Henriques, Jang & Chua, 2009). MdCIP8, for instance, is a RING finger protein that can interact with MdCOP1 to inhibit the accumulation of anthocyanins (Kang et al., 2020). In addition to those already mentioned, this study identified several other MYB-interacting proteins, including the SGT1 protein, which is associated with ubiquitin ligases and potentially plays a broad role in disease resistance (Peart et al., 2002). The relationship between PbrMYB114 and the ubiquitination pathway appears to be particularly intimate, suggesting a significant interaction. Furthermore, the GDT1-like protein 4, acting as a cell membrane calcium transporter, is hypothesized to participate in the facilitation of transmembrane calcium ion (Ca2+) transport (Li et al., 2022b). The precise regulatory influence of these proteins on anthocyanin biosynthesis, however, is not yet established and requires additional investigation. Consequently, forthcoming research endeavors will aim to delineate the intricate regulatory mechanisms and the networks encircling PbrMYB114, which are believed to be pivotal in controlling the synthesis of anthocyanins.

A range of environmental factors (including light and temperature) and plant hormones (including ethylene and auxin) can regulate the biosynthesis of anthocyanins (Ni et al., 2020). The SCFTIR1-Aux/IAA-ARF is the primary mediator of auxin-induced signaling activity in plants (Lavy & Estelle, 2016), and auxin can reportedly inhibit anthocyanin accumulation in apples (Ji et al., 2015a, 2015b). Auxin-related gene families are also reportedly associated with the accumulation of anthocyanins (Wang et al., 2021a). MdIAA26 has been demonstrated to be capable of counteracting auxin-induced suppression of the accumulation of anthocyanins (Wang et al., 2020). Here, PbrARP and PbrSAUR21 were identified as auxin-related proteins capable of interacting with PbrMYB114. Previous studies have demonstrated the negative regulation of the PpARP1 and PpARP2 proteins in pears in response to exogenous auxin application such that they have been suggested to play a role in regulating auxin signal responses in the context of pear fruit development (Shi, Zhang & Chen, 2013). SAUR is an auxin-responsive protein that is also capable of impacting auxin levels, polar auxin transport, and auxin pathway-related gene expression (Xu et al., 2017). In Arabidopsis, the SAUR19 protein exhibited phenotypes associated with the regulation of auxin activity (Spartz et al., 2012). The present results align well with past evidence demonstrating that high auxin levels can inhibit anthocyanin production in apples. Given these findings, PbrARP and PbrSAUR21 may serve as regulators of anthocyanin accumulation through their participation in auxin-induced signal transduction.

PbrMT2 and PbrMT3 were identified as members of the metallothionein (MT) family, both exhibiting the capacity to interact with PbrMYB114. Overexpression of PbrMT2 in ’Duli’ seedlings led to an increased accumulation of anthocyanins, as depicted in Fig. 6A. This outcome supports the hypothesis that PbrMT2 may play a role in the regulation of anthocyanin accumulation, potentially through its regulation of copper (Cu) homeostasis. Metallothioneins (MTs) are cysteine-rich low-molecular-weight metal-binding proteins that serve as key mediators of heavy metal detoxification and the maintenance of intracellular copper Cu(I) and zinc Zn(II) homeostasis (Calvo, Jung & Meloni, 2017). Different MT family proteins exert different functions, with MT2, for example, serving as a Cu-enhanced protein correlated with Cu tissue levels throughout the plant body, whereas MT2 is unrelated to Cu homeostasis (Fidalgo et al., 2013). The transfer of additional Cu ions into the vacuoles from the cytosol may enhance vacuolar anthocyanin stability. Anthocyanin accumulation is closely tied to Cu homeostasis, with relatively high levels of Cu facilitating the expression of genes related to anthocyanin biosynthesis (Hu et al., 2022). However, additional studies will be essential to better clarify the extent to which MT proteins and Cu transport maintain Cu homeostasis and thereby regulate anthocyanin levels, and to unveil the underlying mechanisms.

Conclusions

Our study confirms the regulatory role of PbrMYB114 and its partner PbrbHLH3 in anthocyanin biosynthesis in ‘Yuluxiang’ pears. The identification of PbrMT2 and PbrMT3 as interacting proteins that enhance anthocyanin accumulation provides new insights into the molecular mechanisms controlling fruit pigmentation. This research lays the groundwork for future efforts to improve pear quality through targeted genetic manipulation, with implications for both the fruit industry and consumer health.

Supplemental Information

Supplemental Information 1 Confirm positive interactors.

Supplemental Information 2 The original image.

Supplemental Information 3 RT-qPCR raw data.

Supplemental Information 4 MIQE checklist.

We appreciate the linguistic assistance provided by TopEdit during the preparation of this manuscript.

Additional Information and Declarations

Competing Interests

Author Contributions

Data Availability

The authors declare that they have no competing interests.

Qingwei Liu performed the experiments, prepared figures and/or tables, and approved the final draft.

Ge Gao performed the experiments, prepared figures and/or tables, and approved the final draft.

Chen Shang analyzed the data, prepared figures and/or tables, and approved the final draft.

Tong Li analyzed the data, prepared figures and/or tables, and approved the final draft.

Yadong Wang analyzed the data, prepared figures and/or tables, and approved the final draft.

Liulin Li conceived and designed the experiments, authored or reviewed drafts of the article, and approved the final draft.

Xinxin Feng conceived and designed the experiments, authored or reviewed drafts of the article, and approved the final draft.

The following information was supplied regarding data availability:

The raw data is available in the Supplemental Files.

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
