# Peer review of "Screening and verification of proteins that interact with the anthocyanin-related transcription factor PbrMYB114 in ‘Yuluxiang’ pear"

_PeerJ, doi:10.7717/peerj.17540_

## Round 0.1 · original submission · Major Revisions

· Academic Editor

Major Revisions

Please review the detailed comments and prepare a revised manuscript along with a detailed rebuttal letter.

**Language Note:** The review process has identified that the English language must be improved. PeerJ can provide language editing services - please contact us at [email protected] for pricing (be sure to provide your manuscript number and title). Alternatively, you should make your own arrangements to improve the language quality and provide details in your response letter. – PeerJ Staff

Reviewer 1 ·

Basic reporting

The manuscript was written in clear English, but there are a few things to improve the overall quality of the manuscript which is written in the additional comments. Sufficient field background and context provided.

Experimental design

Valid experimental design and hypothesis testing. The research question is defined and relevant. Methos are sufficient details.

Validity of the findings

The findings are valid and will provide further discussion in the science research community. There are improvements on the figures and legends which will be detailed in the additional comments.

Additional comments

The manuscript was written in clear English, but there are a few things to improve the overall quality of the manuscript which is written in the additional comments.

1. Remove numbering for each sections in the methods and results. Please check the format of other papers published in this journal for reference.

2. line 47: Yuluxiang pears was mentioned without the species name and the full species name only appeared in the methods in line 89. When first mentioned, the pear must be in the full species name.

3. Please add some brief introduction of MYB114 and I am wondering has MYB114 been identified in the Yuluxiang pears before? How about other MYB regulators (activators, repressors). It would be good to expand the introduction with more literature on MYB regulations.

4. type in line 59 Pp4ERF24, please check for these typos elsewhere in the manuscript

5. line 62, space between biosynthesis and reference.

6. Remove numberings in results section.

7. line 175, where is the PcMYB114 in the phylogeny tree?

8. line 183, Fig.2B should be Figure 2B, please use the same in-text referencing, check journal format.

9. There is no mention of Figure 2C in the results.

10. line 246 - 248, redundancy in the sentences, please rephrase.

11. line 285 - 286, mention Cu and Zn in full when first mentioned.

Figure 1. marker size missing on the ladder. include accession number of each protein either in the legend or methods. No mention of how the phylogeny tree was constructed in the methods section.

Figure 2. no explanation of what the a, b, c means. Is it the p-value or the comparison between treatment? There was also no mention of light treatment mentioned in the main text. Figure 2C, move y axis title closer to the axis

Figure 3. What is lane 1 and 2 in A and B? In D, lane 1 is the marker, and sample 1 starts in lane 2. please clearly indicate that in the legend.

·

Basic reporting

The manuscript presents interesting research unveiling a part of anthocyanin biosynthesis mechanism. The study describes the role transcription factor PbrMYB114 and its partner PbrbHLH3 in anthocyanin accumulation in pear fruits. The methodology is sound, but the manuscript has few ambiguities that need to be addressed before it can be considered for publication.
However, language needs improvement. I observed typos throughout the manuscript and thus would recommend the authors to thorough proofread for such typos and grammar mistakes. It would be good, if the authors get it proofed by a fluent English-speaking scientist.
The results and discussion are well described. However, i suggest expanding the discussion section as some of the results are missing and not discussed.
Figures seem good.

Major portion of Abstract contains methodology. I suggest adding some more results (quantified).
In introduction, it would be good to add significance of anthocyanin for the plants as well besides the pigmentation function.
Generally, how many plants were used for “Yulixiang”, as it says pears (plural)? How many replications were used in the experiment, should also be mentioned.
Yuluxiang and Duli are different species of Pear? Or are these cultivar names? How many plants were taken for each species/cultivar?
The Methodology section only mentions the anthocyanin content measurement at L156. The detailed methodology and details on measurement procedures are missing all together.
I could not find specific results related to transient expression assay of ‘Duli’ seedlings (Pyrus betulaefolia)”. Rather, author has generalized as “pear leaves” in the Results and discussion sections. Please clarify.
There should be a list of abbreviations, containing frequently used terminologies that are not explained in the manuscript, for instance 3-AT (3-amino-1,2,4-triazole, I suppose?)

Experimental design

The article is based on anthocyanin content and so is presented in the results section. However, I could not see any procedure for the measurement of anthocyanin. I suggest adding the procedure in methodology and explaining in results and discussion as well.
Additionally, please see following suggestions:
L90: If the sampling orchard is well known place, add name. Otherwise provide coordinates and exact location.
L108: 100 mg·L-1, correct it.
L129: RY8001, ProNet Biotech), add state and country.
L130: (http://peargenome.njau.edu.cn, (accessed on 23 February 2022). Correct formatting
L134: Y2HGold yeast cells underwent transformation. Carefully check for spaces, proof throughout the manuscript.
L154: 24°C under an artificial light source, you mean in a growth/climate chamber? Mention specifics of the instrument in that case.
L155: At 10 days post-infection, leaves were images?? Correct grammar, it does not make sense.
L162: Light Cycler 96 Real-Time PCR System. Isn’t it LightCycler? Add state and country.
Additionally, follow similar formats for all the instruments details as in nucleic acid analyzer (Thermo Nanodrop, DE, US).
L164: It was attempted to utilize the 2-ΔΔCT method to assess relative expression. Provide reference and rephrase the sentence scientifically.

Validity of the findings

Replications need to be clarified.
Data is sound and statistically OK.
L223: Why were only two proteins selected? How was the selection made and on what basis?
It would also be promising to add a figure depicting/presenting the interaction between these 25 proteins. It would also be interesting to know the function/influence of these proteins directly or indirectly to anthocyanin biosynthesis/accumulation. I recommend adding a paragraph to the discussion section.
Formatting of Figure is in-consistent. See L239 and 245, 294, for instance. Correct it throughout the manuscript.
L258: Correct format as per guidelines.
L263-265: The observed interaction between PbrMYB114 and the RING-H2 finger ATL43-like protein in this study may indicate that PbrMYB114 protein is involved in ubiquitin modification… How was this conclusion made. Add explanation and refer to relevant results.
L241: applies? Is it correct?
L292-295 should be at the start of the paragraph, such that the results are presented first and then comes the explanation and supporting literature.
The conclusion section should include a part of conclusive results along with generalized results to present a better picture of the manuscript for the readers.
Figure 3A. What do the 1st and 2nd lane contain? Are these from different plants?
Figure 3B. cDNA of what? Add in caption. Similar for C part of figure. Cell density for what? Should be added in caption.

Additional comments

L30-31 mentions health benefits, but what exactly? Better to mention few directly associated to anthocyanins.
L32-34: too long sentence, difficult to understand. Break it and simplify.
L46: pear fruits(Feng et al. 2015a), correct it
L66: biosynthesis(Li et al. 2022a), correct it
L101: oligonucleotide(dT)
L45-47: Sentence does not make sense, please check and correct.
L59-60 and 68: As focus is MYBs, its TFs should be mentioned first. Also, for general audience, give a background to ERF and COP1, as you presented for MIEL1.

Reviewer 3 ·

Basic reporting

Here, the authors evaluate the possible proteins that putatively interact with PbrMYB114 in Yuluxiang pear to regulate anthocyanin synthesis. Also, they validate the interaction between PbrMYB114 and PbrMT2/PbrMT3 and compare it with anthocyanin production using plant transformation. The results are very interesting and contribute to the knowledge in this field. However, some changes are needed to improve the manuscript.
Minor comments
-Line 81: Please correct Yulixiang
-It’s important to define what PbrMT2 and PbrMT3 are the first time you mention them.
-When you refer to PbrbHLH3 as “the partner”, do you mean that it is paralogous? Please explain.
-The definition of some abbreviations are missing. i.e. HLH, MBW, MIEL, COP1, among others.
-Lines 98-99: Please keep a clear sequence of your explanations. You mention the cDNA kit, but then you explain something about RNA and then come back to cDNA.
-Lines 114, 116, 162: I don’t know if the word “headquartered” is the most appropriate for this purpose. You can mention the city in parentheses.
-Line 163: did you attempt, or did you do? Please add the reference for the formula.
-Lines 174-175: Please indicate from which species.
- Figure 3. Please indicate what “M” and each lane means. How much RNA did you load onto the gel?
-Please consider changing the order in your discussion, to finish it with the MT2 explanation, as these results are part of your validation.
- Figure 4. Please correct PbrMMYB114
- Figure 6. Please indicate which statistical analysis you used.


Major comments
-Please explain why Yuluxiang pear is a good model to study the anthocyanin pathway, considering that it’s poor in anthocyanins.
-Please explain why you select MT2 and MT3 for the validation.
-In your discussion, you mention that your results are very similars to those reported for PyMYB114 in Pyrus bretschneideri. However, you didn’t include this organism in your phylogeny analysis. It will be interesting to compare your organism with this.

Experimental design

Please indicate which statistical analysis you used.

Validity of the findings

No comment

Additional comments

No comment

---

## Round 0.2 · Minor Revisions

· Academic Editor

Minor Revisions

Please address the reviewers' comments and submit a revised version. Please be aware that if language or editing issues remain, they will delay acceptance. Please inquire about the PeerJ editorial proofreading services.

**Language Note:** The Academic Editor has identified that the English language must be improved. PeerJ can provide language editing services - please contact us at [email protected] for pricing (be sure to provide your manuscript number and title). Alternatively, you should make your own arrangements to improve the language quality and provide details in your response letter. – PeerJ Staff

·

Basic reporting

The language is consistent and clear throughout the manuscript.
Literature is sufficiently provided.
Hypothesis may be added at the end of introduction section. L90-95: The last paragraph should also contain hypothesis of the study along with objectives.

Experimental design

The Duli seedlings origin? Same as the Yuluxiang pears?
L179: Replace measures with measurement
Sections “Evaluation of the role of PbrMYB114 as a regulator of pear anthocyanin biosynthesis” and “Construction of a yeast two-hybrid library” should strictly contain results only. Avoid repeating the methodology and merge the methodology parts into relevant sections.
L32, L34 and L70, L148: Add spaces before citations. Such mistakes are common in the article, please proofread the article carefully.
The sampling details explained in response letter should be incorporated into manuscript to clarify the readers.
Abbreviations should be added where 1st used. For instance, 3-AT is first used on L140 but explained later at L236.
I still could not find the reference for the 2−∆∆CT method “The relative expression of the genes was calculated using the 2−∆∆CT method”.
L88: Transcription factor should be abbreviated as TFs following L57.

Validity of the findings

Sections “Evaluation of the role of PbrMYB114 as a regulator of pear anthocyanin biosynthesis” and “Construction of a yeast two-hybrid library” should strictly contain results only. Avoid repeating the methodology and merge the methodology parts into relevant sections.
L222: 1000 to 2000 bp. Some fragments are even around 500bp. Please check and correct.
L285: Correct the citation format.
L311: proteins in pairs or pears?
Reference formatting needs to carefully corrected. For instance; L374-title should be sentence case, what is ARTN?, scientific names are not italicized (Malus sieversii on L395 and L398), L440-check punctuations.

Reviewer 3 ·

Basic reporting

The authors attended some of the comments. Unfortunately, important points were not attended. For example:

-The authors didn't add the reference to 2-ddct formula.
-The statistical analyses are not sufficiently described.
-Please carefully review the nomenclature of genes and proteins.
-In your discussion, you mention that your results are very similar to those reported for PyMYB114 in Pyrus bretschneideri. However, you didn’t include this organism in your phylogeny analysis. It will be interesting to compare your organism with this.

Experimental design

No comment

Validity of the findings

No comment

Additional comments

No comment

---

## Round 0.3 · accepted · Accept

· Academic Editor

Accept

Thanks for your revised version.